# Effects of climate change on the movement of future landfalling Texas tropical cyclones

Pedram Hassanzadeh [1,2 ✉], Chia-Ying Lee[3], Ebrahim Nabizadeh [1], Suzana J. Camargo [3], Ding Ma [4] & Laurence Y. Yeung [2]

The movement of tropical cyclones (TCs), particularly around the time of landfall, can substantially affect the resulting damage. Recently, trends in TC translation speed and the likelihood of stalled TCs such as Harvey have received significant attention, but findings have remained inconclusive. Here, we examine how the June-September steering wind and translation speed of landfalling Texas TCs change in the future under anthropogenic climate change. Using several large-ensemble/multi-model datasets, we find pronounced regional variations in the meridional steering wind response over North America, but——consistently across models——stronger June-September-averaged northward steering winds over Texas. A cluster analysis of daily wind patterns shows more frequent circulation regimes that steer landfalling TCs northward in the future. Downscaling experiments show a 10-percentage-point shift from the slow-moving to the fast-moving end of the translation-speed distribution in the future. Together, these analyses indicate increases in the likelihood of faster-moving landfalling Texas TCs in the late 21st century.

[1] Department of Mechanical Engineering, Rice University, Houston 77004 TX, USA. [2] Department of Earth, Environmental and Planetary Sciences, Rice University, Houston 77004 TX, USA. [3] Lamont-Doherty Earth Observatory, Columbia University, Palisades 10964 NY, USA. [4] Department of Earth and Planetary Sciences, Harvard University, Cambridge 02138 MA, USA. ✉email: pedram@rice.edu

Since the beginning of the 21st century, Texas has experienced a number of devastating TCs; the three costliest ones are Tropical Storm Allison (June, 2001), Hurricane Ike (September, 2008), and Hurricane Harvey (August to September, 2017), which all caused severe damage around Greater Houston, the most populous and industrialized region along the Texas coast. Harvey, Ike, and Allison, with estimated damage of $125B, $34.8B, and $11.8B, are respectively the 2nd, 7th, and 16th costliest TCs affecting the mainland of the United States (US), after accounting for inflation[1]. The critical measures needed to strengthen the resiliency of the Texas coastline, and in particular the Houston-Galveston area, against future TCs require an understanding of the region's past TCs and their potential changes in the future in a changing climate.

The damage from the aforementioned TCs had different characteristics: Harvey and Allison caused flood damage due to record-breaking rainfall[2–6], while Ike caused damage due to strong winds and a record-breaking storm surge[7,8]. Several factors (e.g., intensity, size, landfall angle, translation speed, and sea level) influence the characteristics of the damage from a TC, and various consequences of climate change (e.g., higher sea-surface temperature (SST), increased atmospheric moisture, changes in large-scale circulation) influence these factors[9–12].

Changes in TCs' characteristics such as frequency and intensity have been extensively studied in the past[13–16]. More recently, a number of studies have investigated how the translation speed of TCs at the global or basin-wide scale might have changed in the past few decades, or are projected to change in the future under anthropogenic climate change; however, their findings have remained inconclusive[10,11,17–24].

The focus of this paper is on a specific question: how will climate change influence the movement of future landfalling Texas TCs, and specifically their translation speed? To answer this question, we use several large-ensemble/multi-model datasets and clustering and downscaling techniques. Examining changes in the June to September steering winds and frequency/pattern of clustered daily steering winds shows an increase in the northward meridional steering winds over Texas in the period of 2074–2100 compared to 1979–2005 under the RCP8.5 emission scenario. We suggest that this regionally robust change in atmospheric circulation is associated with an intensifying and west-ward shifting Atlantic subtropical high and a weakening American monsoon. Consistent with these changes in steering winds, downscaling experiments show that in the future, the relative frequency of fast-moving TCs (speed ≥20 km h$^{-1}$) increases by ~6% while that of slow-moving TCs (speed ≤5 km h$^{-1}$) decreases by ~4%, indicating a 10-percentage-point shift from the slow-moving end to the fast-moving end of the translation-speed distribution. Our results do not show any evidence for an increase in the likelihood of slow-moving landfalling Texas TCs in the late 21st century under climate change; on the contrary, our results indicate a higher probability of fast-moving TCs.

## Results

**Large-scale circulation and tracks of most-devastating past Texas TCs.** Figure 1 compares the tracks of Harvey, Allison, and Ike, the rainfall during each storm, and the large-scale

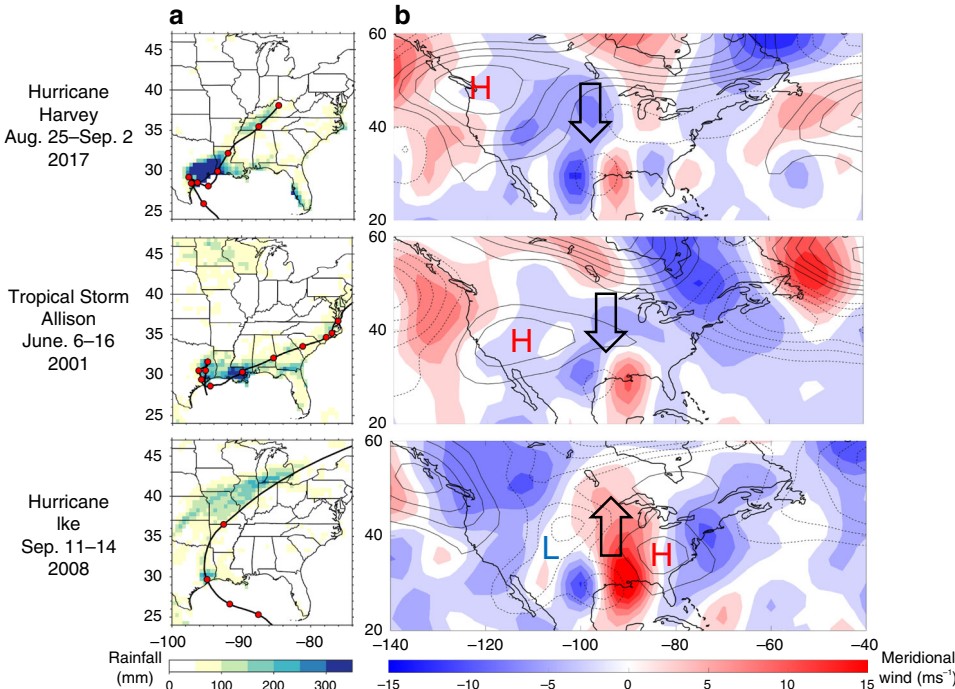

**Fig. 1 Track, rainfall, and large-scale circulation. a** The track of Hurricane Harvey, Tropical Storm Allison, and Hurricane Ike, and the rainfall (shading) during each storm. The red circles mark the position of the center of the storm at 9:00 a.m. UTC during the dates shown for each storm (data from International Best Track Archive for Climate Stewardship, version 4). The shading shows the cumulative rainfall during the time of each storm (Harvey: 25 August–2 September 2017; Allison: 6–18 June 2001; Ike: 8–15 September 2008; data from the National Oceanic and Atmospheric Administration Climate Prediction Center). **b** The anomalous meridional steering wind (shading) and anomalous geopotential height at 500 mb (Z500, contour lines) on the day of Ike's landfall (13 September 2008) and averaged over the day of landfall and the next two days for Harvey (26–28 August 2017) and Allison (6–8 June 2001). Data are from NCEP-DOE reanalysis. Due to Ike's fast movement, the large-scale circulation is plotted only on the day of landfall. The anomalies are computed with respect to a 15-day (17-day) running mean around the landfall date (landfall date +1) for Ike (Harvey and Allison) for 1979–2018, with the landfall year excluded. The interval of contour lines is 25 m and continuous (broken) lines show positive (negative) values. High-pressure (H) and low-pressure (L) systems are marked. See Methods for further details.

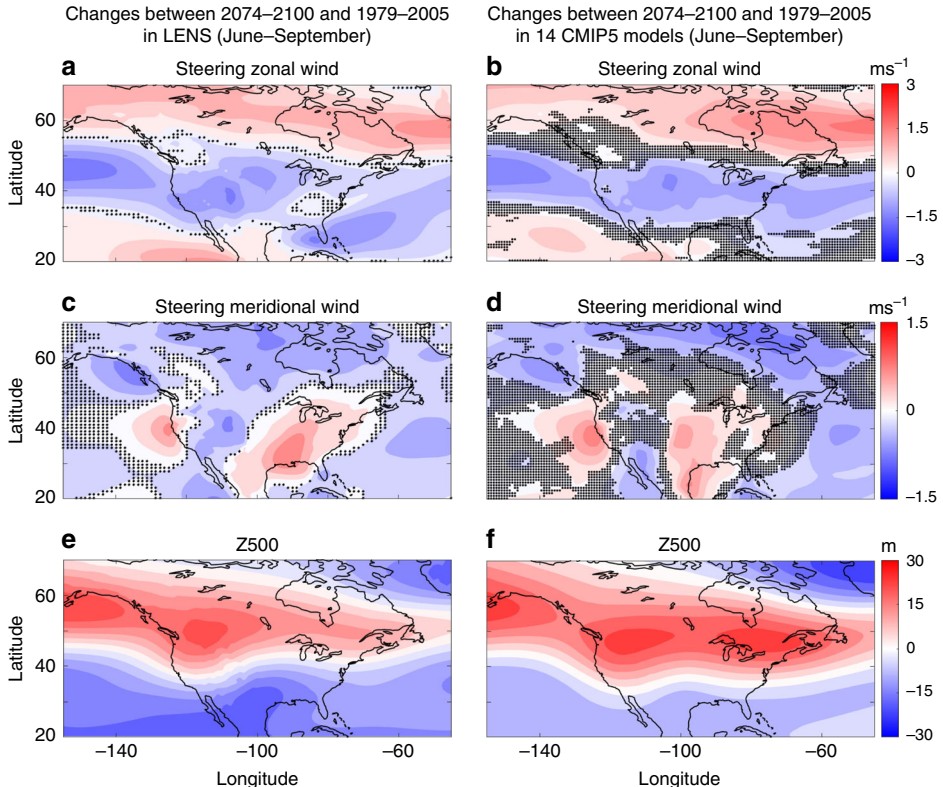

**Fig. 2 Changes in the large-scale circulation and steering winds under climate change.** Changes are computed as June to September averages in the period of 2074–2100 minus those in the period of 1979–2005. **a**, **c**, **e**: Using 40 ensemble members of National Center for Atmospheric Research's Large Ensemble Community Project (LENS). Stars show where the difference is not statistically significant, based on a two-tailed *t* test at 95% level. A domain-averaged increase of 112.6 m is removed from geopotential height at 500 mb (Z500) for better illustration. **b**, **d**, **f**: Using multi-model-mean from 14 Coupled Model Intercomparison Project 5 (CMIP5) models. Dots show where fewer than 10 models (out of 14) agree on the sign of the change. A domain-averaged increase of 122.7 m is removed from Z500. See Methods for further details. See Supplementary Fig. 1 for the projected changes using 100 ensemble members of Max Planck Institute for Meteorology Grand Ensemble (MPI-GE), and using 20 ensemble members of Geophysical Fluid Dynamics Laboratory Large Ensemble (GFDL-LE). Supplementary Figs. 2–9 show similar plots but separately for each month of June to September.

atmospheric circulation over North America around the time of their landfall. Both Harvey and Allison stalled over southeast Texas for around 5 days, which was a major contributor to the substantial rainfall over Houston[2,3]. Around the time of each storm's landfall, an anomalous high-pressure system existed over western US, which caused anomalous southward steering winds, slowing down or stopping the storm from moving north and reaching the midlatitude westerlies (following Lee at el.[25], steering winds are defined as the weighted average of lower-level 850 mb (80%) and upper-level 200 mb (20%) winds). Ike, however, was a fast-moving TC that crossed Texas in less than one day. Around the time of Ike's landfall, an anomalous high-pressure system was present over eastern US and an anomalous low-pressure system was present over mid-western US, which together resulted in strong anomalous northward steering winds, leading to Ike's fast northward translation speed. These examples show, to the leading order, the effects of the large-scale circulation on the movement of the three most-devastating Texas TCs in the 21st century.

**Future changes in June-to-September-averaged steering winds.** How, then, will the large-scale circulation and steering winds in this region respond to anthropogenic climate change? Projections of regional changes in atmospheric circulation are known to often have large uncertainties, in particular due to natural variability and model biases[26]. To reduce these uncertainties, we use three sets of large-ensemble simulations with 20, 40, and 100 members, as well as single-member simulations from 14 Coupled Model

Intercomparison Project Phase 5 (CMIP5) models (see Supplementary Table 1). Figure 2 shows the projected change in June to September-averaged zonal and meridional steering winds and geopotential height at 500 mb (Z500) using the Community Earth System Large Ensemble Project (LENS) dataset and the multi-model-mean of CMIP5 simulations. Supplementary Fig. 1 shows the projected changes in the two other large-ensemble datasets – the Max Planck Institute for Meteorology Grand Ensemble (MPI-GE) and the Geophysical Fluid Dynamics Laboratory Large Ensemble (GFDL-LE).

Consistently across all models, the steering eastward winds are projected to decline between 30°N and 50°N, associated with the poleward shift of the midlatitude westerlies[27]. All models predict a robust northward steering wind response over Texas that typically extends from the south of 30°N to the north of 40°N. This northward wind response is often (but not in all cases) part of a dipolar meridional wind pattern with a southward component west of ~100°W; see Fig. 2c, d and Supplementary Fig. 1c, d. Note that the magnitude of this northward response is comparable to the climatology, e.g., using Houston (95°W, 30°N) as a reference, the response in LENS is ~33% of the June-to-September-averaged meridional steering wind.

This robust northward steering wind response over Texas could potentially lead to an increase in the northward translation speed of landfalling Texas TCs, while the weakening of the westerlies (statistically significant north of 30°N–35°N depending on the model, see Fig. 2 and Supplementary Fig. 1) could cause a decrease in the eastward translation speed of the TCs once they

reach the midlatitudes (see the Discussion and recent papers by Yamaguchi et al.[23] and Zhang et al.[24] for the implications of these changes). The northward steering wind response adds to the natural tendency of TCs to move poleward due to beta advection[28,29] and is robust from June to September (see Supplementary Figs. 2–9). In most model projections, this northward wind response is concentrated over a relatively narrow band over Texas. Examining the model projections in Z500, however, shows some differences across models. In the CMIP5 multi-model mean, there are two distinct positive peaks in Z500 response over the northwest and northeast US (Fig. 2f). In contrast, the LENS (Fig. 2e) and GFDL-LE (Supplementary Fig. 1e) models show only one peak, in the northwest and northeast, respectively, while MPI-GE projects a broad increase of Z500 over Canada (Supplementary Fig. 2f).

The high-pressure responses over northwest US resemble the anomalous high-pressure system during Harvey and to some extent Allison (Fig. 1). However, the changes in meridional steering winds over Texas and the midwest in Fig. 1 (during Harvey and Allison) and in Fig. 2e, f (by the end of the 21st century) are opposite. In LENS (Fig. 2e), there is a low-pressure (cyclonic) response centered around (100°W, 30°N) that is consistent with the northward wind response over Texas; however, such low-pressure response is missing from, or is much less pronounced in, the other models. These results suggest that understanding the source(s) and underlying mechanism of the northward steering wind response over Texas might require a closer examination of the subtropical processes and additional variables.

**Mechanism of June-to-September-averaged steering wind changes.** Recently, Wills et al.[30] investigated the response of the northern hemisphere stationary waves to climate change. Their analysis of the historical and RCP8.5 simulations of CMIP5 models show a similar narrow band of northward wind over Texas in the stationary components of both low- and upper-level meridional winds from June to August (see their Fig. 2a, d). Following their analysis, we examine the responses of the stationary components of June to September 850 mb and 200 mb streamfunctions, as well as sea-level pressure, in the LENS, GFDL-LE, and MPI-GE datasets (Supplementary Fig. 10). The increase in the low-level northward wind over Texas is consistent with the intensification and westward expansion of the North Atlantic subtropical high (see Supplementary Fig. 10a–f; also see Fig. 2a, c of Wills et al.[30]), which predominantly results from the enhanced land-ocean thermal contrast[31]. Furthermore, the increase in the upper-level northward wind over Texas is consistent with the weakening of the North American monsoon[32,33] (see Supplementary Fig. 10g–i; also see Fig. 2b of Wills et al.[30]), which is due to increases in static stability with sea-surface warming[32]. These changes in the Atlantic subtropical high and American monsoon constructively influence the low- and upper-level meridional winds over Texas, leading to a remarkably robust increase in the northward steering winds.

**Future changes in daily steering wind regimes.** In the above analysis, we discussed changes in the June to September and monthly-mean steering winds. Below, we will examine how climate change affects the daily steering wind patterns over southern US by applying a self-organizing map (SOM) cluster analysis to the LENS daily steering wind vectors (consisting of the zonal and meridional components) in the current (1979–2005) and future (2074–2100) climates (see Methods section). We focus on LENS, which reproduces the 1979–2005 daily steering wind regimes in the NCEP-DOE reanalysis fairly well (see Supplementary Figs. 11 and 12). To examine and quantify changes in

steering wind regimes in the future, in the analysis that is presented hereafter, following the framework of Gervais et al.[34], we apply the SOM analysis to the data of current and future climates combined together (see Methods section). Supplementary Fig. 13 shows the distinct steering wind regimes over Texas: Clusters C1-C5 and C10 have strong southerlies, clusters C6 and C9 have strong northerlies, and clusters C7 and C8 have weak steering winds. Climate change might affect these regimes by changing the frequency or the pattern of each cluster. We emphasize again that each cluster contains days from both current and future periods, and within each cluster, the frequency and pattern might change between the two periods under climate change. Such changes can be quantified and visualized separately (see Eqs. (4) and (5) in Methods and Supplementary Figs. 14 and 15) or together (see Eq. (3) and Supplementary Fig. 16).

Figure 3 shows the results for the five clusters that experience the largest changes between the current and future climates. The northward steering winds associated with clusters C1, C4, and C5 over Texas (Fig. 3a–c) become stronger in the future (Fig. 3d–f), mainly due to increases in the frequency of these clusters (Supplementary Fig. 14). The southward steering winds associated with clusters C6 and C9 (Fig. 3g, h) become weaker in the future, as the changes are northward (Fig. 3I, j). These trends are due to changes in both the frequency and pattern (Supplementary Figs. 14 and 15). Overall, the changes in the two clusters with southward steering winds (C6 and C9) indicate a weakening of such winds (Fig. 3m), and the changes in all the other clusters together indicate a strengthening of the northward steering winds over Texas (Fig. 3k). The total changes across all clusters point to an increase in northward steering winds (Fig. 3n), which suggests an increase in the northward translation speed of landfalling Texas TCs. As expected, the wind pattern shown in Fig. 3n is approximately the same as the combination of the responses shown in Fig. 2a, c. The cluster analysis has further shown that this June-to-September-averaged response arises from an increase, by ~7%, in the daily frequency of regimes that have northward steering winds, and a decrease, by ~7%, in the daily frequency of regimes that have southward steering winds (as well as some changes in the patterns for the latter); see Supplementary Fig. 14.

**TC-CMIP5 downscaling experiments.** Our interpretation of how changes in the daily steering wind patterns affect the movement of future TCs involves the assumption that TCs approach the Texas coastline with the same probability across different clusters. However, both the TC probability and the occurrence of a given cluster are connected to the large-scale atmospheric circulation, in particular the subtropical circulation, which is responsible for moving TCs from their genesis region (e.g., the Atlantic Ocean and Gulf of Mexico) toward the Texas coast. Therefore, next we examine changes in the translation speed of landfalling Texas TCs in synthetic TCs generated by a downscaling model, the Columbia TC HAZard model (CHAZ, Lee et al.[25]). These experiments account for not only the effects of changes in the steering winds and TC frequency, but also changes in other environmental variables including the potential intensity (PI), mid- to low-level moisture, and low-level vorticity. The CHAZ model is downscaled from six CMIP5 models (see Methods section). Statistics of the synthetic TCs generated by CHAZ at the global and basin-wide scales are described in Lee et al.[35]. Here we use only a subset of storms that pass through an area within 300 km from Houston at the historical (HIST, 1981–2005) and late 21st century (RCP8.5, 2071–2099) periods.

Figure 4 shows that the observed landfalling Texas TCs tend to move northward and westward with their translation speed peaking between 5 and 15 km h$^{-1}$. The simulated Texas TCs at

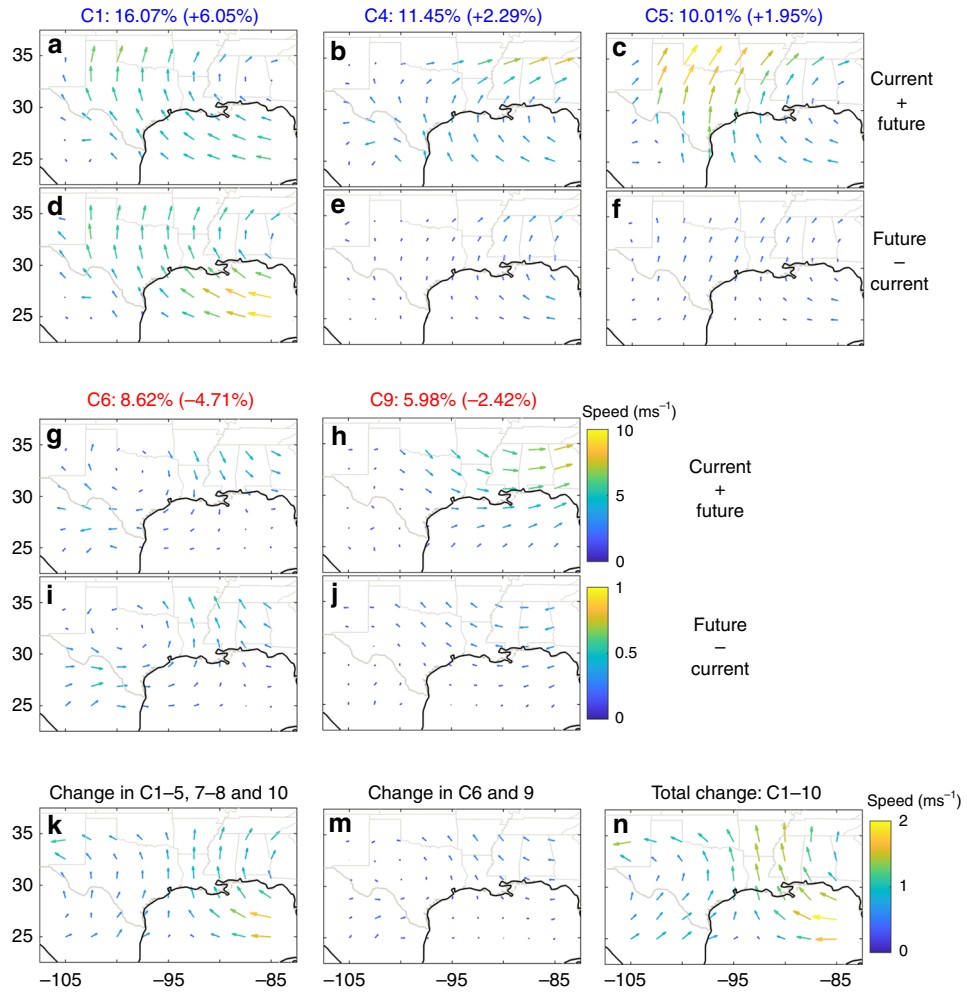

**Fig. 3 Changes in daily steering wind regimes between current and future climates in the Large Ensemble Community Project (LENS) dataset.** Vectors of daily June to September steering wind are classified into 10 clusters using the self-organizing map (SOM) analysis applied to the current (1979–2005) and future (2074–2100 under RCP8.5) periods combined together. All cluster centers and their frequencies are shown in Supplementary Fig. 13. For each cluster $i$, the frequency $f_i^C$ ($f_i^F$) is defined as the number of days in the current (future) period in that cluster divided by the total number of days in the current (= future) period. Panels **a–c** and **g**, **h** show the cluster centers and their frequency, $\bar{f}_i = (f_i^C + f_i^F)/2$, for the five clusters with the largest changes between the future and current climates. Numbers in parentheses are the change in the frequency of patterns within each cluster ($f_i^F - f_i^C$; see Supplementary Fig. 14). Clusters C1, C4, and C5 (C6 and C9) have strong northward (southward) steering winds over Texas. Panels **d–f** and **i**, **j** show the change within each cluster due to change in frequency and change in pattern (future minus current; see Supplementary Fig. 16). **k** (**m**) shows the change, future minus current, in clusters with weak or northward (southward) steering winds over Texas. Panel (**n**) shows the change across all clusters, i.e., the sum of the changes in panels **k** and **m**, which is approximately the change in June to September-averaged steering wind vectors. Panels **a–c** and **g–h**, panels **d–f** and **i**, **j**, and panels **k–n** have the same colorbars. The arrow size and colormap of panels **d–f** and **i**, **j** (panels **k–n**) are scaled such that their values are 1/ 10 (1/5) of those in panels **a–c** and **g–h**.

the HIST period have similar characteristics in both translation speed and direction, although the CHAZ-CMIP5 model underestimates the probability of slow-moving storms (translation speed ≤5 km h$^{-1}$) while overestimating the probability of the fast-moving ones (translation speed ≥20 km h$^{-1}$). Thus, we also analyze data in which this bias has been corrected using two different methods (see below). The probability density function (PDF) in the translation speeds shifts under RCP8.5 toward higher speeds (Fig. 4a): The relative probability of TCs with speed ≥20 km h$^{-1}$ *increases* from 31.4% to 37.6% of all TCs whereas the probability of TCs with speed ≤5 km h$^{-1}$ decreases from 20.4% to 16.6% of all TCs, indicating a 10-percentage-point shift from the slow-moving end to the fast-moving end of the PDF. Analyzing the bias-corrected PDFs leads to the same conclusion (Supplementary Fig. 17). The shift toward fast-moving TCs is mainly due to a shift of the meridional translation speed's PDF toward faster northward speeds (see Fig. 4b and the caption). Results from

downscaling experiments are consistent with those from large-scale circulation analyses discussed earlier.

## Discussion

In this paper we investigate how anthropogenic climate change might affect the steering winds and translation speed of landfalling Texas TC by the end of the 21st century. To reduce the effects of natural variability and model bias, we use the outputs of three sets of large-ensemble simulations and 14 CMIP5 models. We use three different analysis techniques, examining changes in June to September or monthly-mean steering winds over North America, changes in clustered steering wind patterns over Texas, and downscaling experiments for TCs making landfall along the Texas coast (centered on Houston) from six CMIP5 models.

Our results show no evidence for an increase in the probability of slow-moving landfalling Texas TCs by the end of the 21st century.

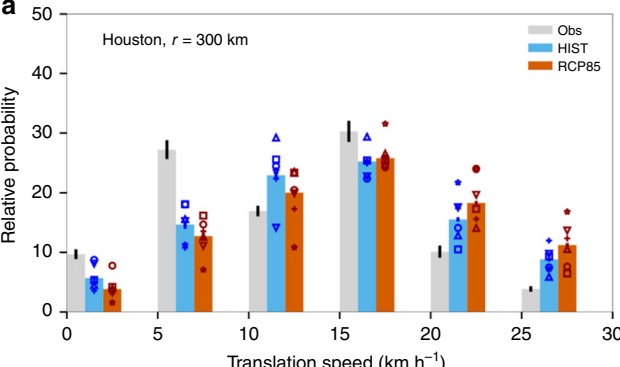

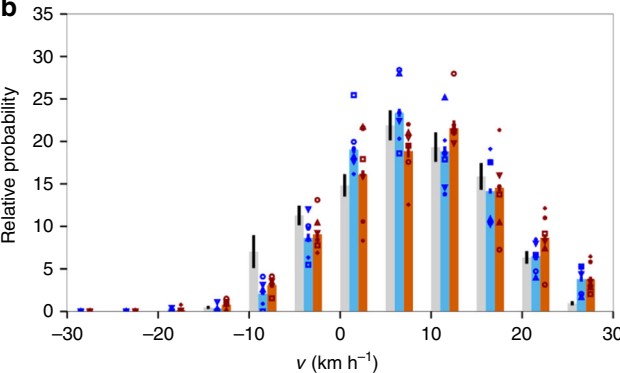

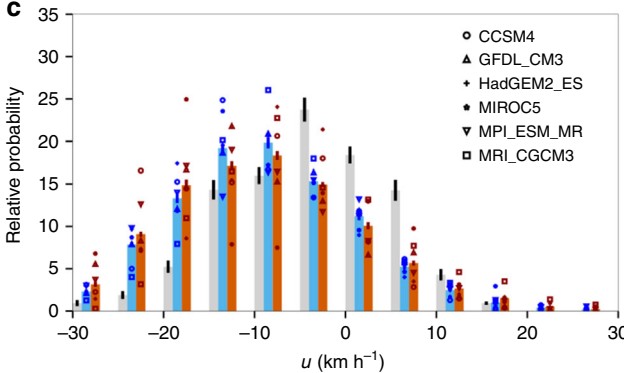

**Fig. 4 Relative probability of the translation speed of tropical cyclones (TCs) that pass through an area within 300 km from Houston. a** Total speed; **b** Meridional speed $v$; **c** Zonal speed $u$. Data from observations (1981–2018, gray), and from the HIST (1981–2005; blue) and RCP8.5 (2071–2099; red) Columbia TC HAZard-Coupled Model Intercomparison Project 5 (CHAZ-CMIP5) downscaling simulations. The observational data are from International Best Track Archive for Climate Stewardship, version 4. The black vertical lines in gray bars show one standard deviation from observations while the symbols along the blue and red bars show the mean values from individual CMIP5 models. See Methods for further details. In **a**, there is ~10% shift from the relative probability of slow-moving TCs toward that of fast-moving TCs under RCP8.5 (see text). In **b**, there is ~9.8% shift from the relative probability of slow-moving TCs ($|v| \leq 5$ km h$^{-1}$) toward that of fast-moving TCs (northward speed $v \geq 15$ km h$^{-1}$) under RCP8.5. In **c**, there is ~2.9% shift from the relative probability of slow-moving TCs ($|u| \leq 5$ km h$^{-1}$) toward that of fast-moving TCs (westward speed $u \geq 15$ km h$^{-1}$). Supplementary Fig. 17 shows the same analysis but with bias correction.

Instead, we find a robust projected increase in the northward steering winds over Texas, which could lead to an increase in the frequency of fast-moving landfalling TCs. Indeed, the downscaling experiments show a decrease in the likelihood of slow-moving TCs

making landfall around Houston, and an increase in the likelihood of the fast-moving ones. The aforementioned changes in the steering winds over Texas appear to be associated with changes in the Atlantic subtropical high and American monsoon. Our results highlight the importance of conducting regional analyses to investigate the effect of climate change on the movement of future TCs, as there are prominent regional variations in the steering winds' future changes. Note that while we focus on Texas in this paper, our multi-model, multi-faceted approach can be readily applied to other regions in future work.

We emphasize that the focus of our study is on potential changes in the movement of landfalling Texas TCs by the end of the 21st century compared to the late 20th century due to anthropogenic climate forcing. This emphasis should be kept in mind when comparing our conclusions with those of other recent studies. For example, Kossin[17] has reported a global slowdown of TCs in the period of 1949–2016, including by 16% over land areas affected by the North Atlantic TCs. However, the analysis of Kossin[17] (and Hall and Kossin[19]) differs in region and period from our results; thus a direct comparison is not valid. Furthermore, some recent studies[20–23] have challenged the findings of Kossin[17]. In particular, Yamaguchi et al.[23] showed that large-ensemble, high-resolution simulations using an atmospheric general circulation model (AGMC) do not support a slowdown of northern hemisphere TCs from 1951 to 2011, and attributed the slowdown reported in Kossin[17] to inhomogeneities of the observational data in the pre- and post-satellite eras.

To examine the effect of climate change, Yamaguchi et al.[23] and Zhang et al.[24] have conducted AGCM simulations with +4 K warmer global SST and compared the TC translation speeds in the period of 2051–2110 versus 1951–2011. Both studies found a decrease in the average TC translation speed at higher latitudes in the future (consistent with the projected decreases in the mid-latitude eastward steering winds in Fig. 1 and Supplementary Figs. 1–9). Yamaguchi et al.[23] also reported an increase in the relative frequency of TCs at higher latitudes. Because the translation speed is much larger in the midlatitudes (due to the westerlies), Yamaguchi et al.[23], as discussed in detail in their paper, found, overall, an increase in the annual-mean global TC translation speed, including a statistically significant increase from 22.1 km h$^{-1}$ to 22.6 km h$^{-1}$ in the North Atlantic basin. While the conclusion of their work appears to be similar to ours, the underlying reasons are entirely different, because of our focus on the Texas coastline (low latitudes) and their focus on hemispheric and basin-wide changes (low and high latitudes). We found an increase in the likelihood of fast-moving landfalling Texas TCs due to the increase in northward meridional steering winds over the northern Gulf region and Texas. Thus, our work and that of Yamaguchi et al.[23] and Zhang et al.[24] answer different, although complementary, questions.

Gutmann et al.[18] have performed pseudo-global warming simulations using the Weather Research and Forecasting (WRF) model and examined how the movement of eastern and south-eastern US TCs might change under RCP8.5 by the end of the 21st century. Among the 22 TCs they examined that had small track changes between the current and future simulations, Ike was the only Texas hurricane (out of two, the other being Rita) with statistically significant change in its translation speed, which decreased from 8.1 m s$^{-1}$ to 6.7 m s$^{-1}$. Our study and that of Gutmann et al.[18] focus on similar periods and region; however, there are major differences in our methodologies. In our multi-model, multi-faceted approach, the conclusions are based on consistent statistical changes of steering winds and synthetic TCs, while Gutmann et al.[18] drew their conclusions from changes in the single-model simulations of a few historical TCs in pseudo-global warming experiments. Thus, changes in the landfalling TC

frequency and impacting angle are not considered in their study. Furthermore, we focus on the movement and translation speed around the time of landfall, while Gutmann et al.[18] considered the average translation speed over the entire lifetime of the storms, e.g., in the case of Ike, from the Caribbean Sea to northern US. Further investigations are needed to fully understand the differences and reconcile the conflicting conclusions between our work and that of Gutmann et al.[18]

Hurricane Harvey's slow movement and stall over Texas was a major contributor to its extreme rainfall and the resulting flooding[2,3,6]. However, the increased possibility of fast-moving landfalling Texas TCs in the future does not necessarily suggest a reduced risk to human life, infrastructure, and ecosystems. For example, the financial damage from Hurricane Ike, a fast-moving hurricane, was comparable to that of the slow-moving Hurricane Harvey. However, as mentioned earlier, the damages were due to differing factors: intense rainfall (which led to extensive flooding) in the case of Harvey, and storm surge-induced flooding and wind gust in the case of Ike. Understanding the main driver(s) of damage by future TCs in each region is crucial for adaptation and mitigation efforts, as different drivers require different——and often costly and controversial——protective measures and strategies (e.g., seawalls for storm surge vs. improved reservoirs and bayou systems for rainfall-induced flooding)[36–39].

The damage from a TC depends on many factors, and to fully assess the risk of future TCs, in addition to changes in their movement, changes in TC size and intensity, as well as sea level, SST, air moisture content, other environmental factors, and even urbanization[40] should also be considered. Some recent studies have quantified the influence of climate change on Hurricane Harvey's and future Texas TCs' rainfall[3–6,10,41] as well as future TC-induced flooding[42]. Our work suggests that further investigation, particularly aimed at disentangling the contributions from changes in dynamics (large-scale circulation) and thermodynamics (temperature) are needed to better understand and constrain the impact of climate change on the risk of TCs making landfall in Texas. Finally, in this paper, as in most other studies, we focus on changes in the late 21st century under the high-emission scenario, RCP8.5 (thus allowing comparison with previously reported results). However, to better inform the adaptation and mitigation efforts, mainly about the time of emergence and the magnitude of these changes in TC movement, similar analyses for the mid-21st century and under other emission scenarios such as RCP4.5 (as e.g., discussed in Knutson et al.[11,14]) should be conducted in future work.

## Methods

**Rainfall and tropical cyclones' track data**. Cumulative precipitation during each storm is calculated for the specified time period using the National Oceanic and Atmospheric Administration (NOAA) Climate Prediction Center (CPC) global unified gauge-based analysis of daily precipitation[43]. The data are plotted at the native resolution of 0.5° × 0.5° in Fig. 1.

The TC track data are obtained from the International Best Track Archive for Climate Stewardship (IBTrACS), version 4[44], which contains interpolated ~3-hourly data for storm positions. Data are overlaid using the Climate Data Toolbox for MATLAB[45].

**NCEP-DOE reanalysis data**. We use 1979–2018 daily averaged zonal and meridional winds ($u$ and $v$) and Z500 from NCEP-DOE reanalysis 2 dataset[46]. In Fig. 1 and the rest of the paper, components of the steering winds are defined following Lee at el.[25] as the weighted average between winds at 850 mb and 200 mb:

$$u_{\text{steering}} = 0.8 \times u_{850} + 0.2 \times u_{200} \tag{1}$$

$$v_{\text{steering}} = 0.8 \times v_{850} + 0.2 \times v_{200} \tag{2}$$

**Large-ensemble datasets**. We use monthly averaged $u$, $v$, Z500, and sea-level pressure from three large-ensemble datasets: The National Center for Atmospheric Research (NCAR) Community Earth System Model (CESM) Large Ensemble Community Project (LENS)[47], the Max Planck Institute for Meteorology (MPI)

Grand Ensemble (MPI-GE)[48], and the Geophysical Fluid Dynamics Laboratory (GFDL) Climate Model version 3 (CM3) Large Ensemble (GFDL-LE)[49]. We also use daily averaged $u$ and $v$ from LENS and GFDL-LE.

These datasets contain data from fully coupled atmosphere-land-ocean-ice simulations of the period 1920–2005, based on the historical radiative forcing, and the period 2006–2100, where the forcing is chosen based on the high-emission scenario RCP8.5. For each period, an ensemble with 20 (GFDL-LE), 40 (LENS), or 100 (MPI-GE) members are simulated by starting from initial conditions that differ in small random perturbations. The LENS, MPI-GE, and GFDL-LE models have horizontal resolutions of approximately 1°, 1.8°, and 2°, respectively.

To compute mean changes in large-scale circulation in the large-ensemble and CMIP5 datasets (Fig. 2 and Supplementary Figs. 1–9), we analyze and compare data from 1979–2005 (referred to as "current" climate) and 2074–2100 (referred to as "future" climate).

**CMIP5 datasets**. We use monthly averaged $u$, $v$, and Z500 from 14 models in Phase 5 of the Coupled Model Intercomparison Project (CMIP5)[50]; see Supplementary Table 1. We use the historical and RCP8.5 simulations of each model. To compute the multi-model-mean, we first interpolate data from all models to the highest horizontal resolution (0.75° × 0.75°), and then calculate the average over all models.

**Cluster analysis**. To cluster daily steering wind patterns and identify the dominant regimes, we use self-organizing map (SOM)[51], which is an artificial neural network that has been extensively used to classify climate data[34,52–57]. To conduct the cluster analysis, we compute vectors consisting of the zonal and meridional components of the daily steering winds (Eqs. (1)–(2)) for each grid point inside a box around Texas, and then apply the SOM algorithm on these daily vectors. Note that we do not conduct any pre-processing on the daily vectors before applying the SOM algorithm. For SOM, we use the selforgmap subroutine in MATLAB's Deep Learning Toolbox. Different layers of neurons with 1000 ordering phase steps are set to classify wind patterns. The distance between layers is calculated using the linkdist function and the layer topology function is set to be hextop, which creates hexagonal patterns for neurons.

In order to find the uncertainties of the frequencies, we perform 50 repetitions of SOM clustering, each time using one fifth of the total available data chosen randomly. Pattern correlation is used to find the correspondence between the cluster centers obtained each time and those obtained from the entire dataset. In Supplementary Figs. 12–14, the uncertainties of the frequencies are reported as standard error.

We focus on the LENS dataset, which provides daily wind data for a 40-member ensemble for each period, and in the period of 1979–2005, reproduces the June–September steering wind clusters of reanalysis data fairly well (compare Supplementary Figs. 11 and 12 and see their captions). Note that the results of these two figures are obtained by conducting two separate SOM analyses: One applied to the daily steering wind vectors in the reanalysis data (Supplementary Fig. 11) and one applied to the daily steering wind vectors in the LENS current climate data (Supplementary Fig. 12).

Choosing the appropriate number of clusters is a challenging task in using any unsupervised cluster analysis technique. Here, we use the subjective criterion that the number of clusters should be small enough so that the resulting cluster centers have distinctly different wind patterns, yet large enough such that increasing the number of clusters do not lead to new distinctly different patterns. We use 10 clusters (SOM size = 2 × 5); using 12 clusters (SOM size = 3 × 4) leads to the same conclusion.

To investigate how the steering wind patterns change between the current and future climates, following previous studies[34,53,54], we apply the cluster analysis on a dataset consisting of both current and future climates. Here, we follow the framework of Gervais et al.[34]. For each cluster $i$, we define the frequency $f_i^C$ ($f_i^F$) as the number of days in the current (future) period in that cluster divided by the total number of days in the current period, which is equal to the total number of days in the future period. Hereafter, superscripts F and C refer to the future and current climates, respectively. Supplementary Fig. 13 shows the frequency $\bar{f}_i = (f_i^C + f_i^F)/2$ and the center of each cluster $i$ (the cluster center, $\bar{P}_i$, is the average of the daily patterns within each clusters). There are clearly distinct cluster centers, corresponding to regimes involving strong northward (C1, C3-C5, C10) or southward (C6 and C9) steering winds. Then, we analyze, how each cluster changes between the current and future climates, and further quantify, separately, how the change in frequency and change in wind pattern within each cluster contribute to the total change:

Total change in cluster

$$i = f_i^F \times P_i^F - f_i^C \times P_i^C = \bar{P}_i \times \Delta f_i + \bar{f}_i \times \Delta P_i \tag{3}$$

The first term after the second equal sign is calculated as

$$\bar{P}_i \times \Delta f_i = \left(P_i^F + P_i^C\right)/2 \times \left(f_i^F - f_i^C\right) \tag{4}$$

and shows the effect of the change in the frequency of that cluster in the future climate compared to the current climate. Supplementary Figure 14 shows $\bar{P}_i \times \Delta f_i$ from Eq. (4) for each cluster. The second term after the second equal sign is calculated as

$$\bar{f}_i \times \Delta P_i = \left(f_i^F + f_i^C\right)/2 \times \left(P_i^F - P_i^C\right) \tag{5}$$

and shows the effect of the change in the wind patterns in that cluster in the future

climate compared to the current climate. Supplementary Figure 15 shows Eq. (5) for each cluster. See Gervais et al.[34] for further discussions of the methodology. Supplementary Figure 16 shows the total change (Eq. (3)) in each cluster.

**CHAZ-CMIP downscaling experiments**. CHAZ, the Columbia (TC) HAZard model, is used for generating synthetic storms in the historical period (1981–2005; HIST) and the late 21st century under RCP8.5 (2071–2099; RCP8.5). Lee et al.[25] developed and tested the CHAZ model using recent historical, observation-based reanalysis data. Lee et al.[35] then downscaled the CHAZ model from six CMIP5 models (see Supplementary Table 1) to examine the impact of a warming climate on the global and basin-wide TC activity. Details of the methods and the configuration of the CHAZ model, as well as the CMIP5 models, are described in Lee et al.[25,35]. Below we provide a summary.

The CHAZ model consists of three separate models: genesis, track, and intensity. The genesis model seeds the domain with weak vortices using a seeding rate that depends on environmental conditions through a TC genesis index (TCGI)[58,59]. The track model then moves seeds forward by advection of the environmental steering wind (Eqs. (1)–(2)) plus a beta drift component[60]. The evolution of the storms' intensity is then determined by the intensity model[61,62] using the surrounding large-scale environment via an empirical multiple linear regression model, plus a stochastic component. The stochastic component accounts for the internal storm dynamics and does not depend explicitly on the environment. Intensity at landfall and shortly afterward is calculated from a separate regression model that takes into account both the proximity to land and the environmental conditions. Ambient environmental variables required by the CHAZ model are potential intensity (PI)[63], deep-layer (850 mb to 250 mb) vertical wind shear, the moisture variables – column integral relative humidity (CRH) or saturation deficit (SD), the absolute vorticity at 850 mb, and the steering flow. SD is the difference between the column integrated water vapor and the same quantity at saturation, and the CRH is their ratio. Both are calculated following Bretherton et al.[64].

In this study, we use a subset of synthetic storms from Lee et al.[35] that affect Texas. In Lee et al.[35], the CHAZ model is downscaled from monthly averaged data of six CMIP5 models; see Supplementary Table 1. As discussed in Lee et al.[35], the future projections of the annual frequency of TCs globally and in the North Atlantic region are sensitive to the choice of CRH and SD in TCGI. Consequently, there is a large uncertainty in assessing the frequency of TCs affecting Texas in future climate. However, Lee et al.[35] noticed that the projected changes in the forward speed are not sensitive to the annual TC frequency. Therefore, as our focus is on the changes in the steering winds and the relative probability distribution of the forward speed, we use all the Texas storms from both CRH and SD experiments described in Lee et al.[35]. We refer to synthetic storms from the historical period (1981–2005) as 'HIST' while those from the late 21st century (2071–2099) as 'RCP8.5'.

The CHAZ results from the HIST period are compared to the observations from IBTrACS version 4[44]. We use the 6-hourly storm location (in longitude and latitude) and maximum wind speed from 1981 to 2018. As noted in the Results Section, CHAZ captures the climatology of the observed forward speed of Texas storms relatively well (Fig. 4). Nevertheless, it is noticeable that CHAZ at the HIST period underestimates the probability of slow-moving storms and overestimates the probability of the fast-moving ones. Although our interest is in the differences in the synthetic storms' forward speed in the RCP8.5 period compared to those in the HIST period, it is also reasonable to calibrate the model results to match with observations, i.e., conduct bias correction[42,65]. When doing so, we should examine whether our results and conclusions are sensitive to the bias correction approach. Thus, we conduct additional sets of analyses using bias-corrected data with two distribution mapping approaches applied.

In the first approach, we assume that the relative probability distributions of the forward speed and direction are Gaussian, and we correct the location and scale parameters of the modeled distribution. In other words, the corrected distribution has the same mean and standard deviation as the observed one. In the second approach, we use the quantile-matching technique[66], which allows the CHAZ's distribution to match the entire observed distribution. We derive the correction factors using CHAZ HIST simulations and then apply the same corrections to simulations for RCP8.5. Supplementary Figure 17 shows the relative probability distributions of the translation speed, and its meridional and zonal components, with the Gaussian bias correction method applied. In the caption, the numbers associated with the shifts in the relative probability distributions with Gaussian or quantile-matching bias-correction method are reported. Analyzing the bias-corrected distribution from either approach yields the same conclusion as the one reached with the original data (Fig. 4): a decrease in the relative probability of slow-moving TCs and an increase in the relative probability of fast-moving TCs by the end of the 21st century under RCP8.5 (Supplementary Fig. 17).

## Data availability
The daily precipitation data are available at https://psl.noaa.gov/data/gridded/data.cpc.globalprecip.html. The IBTrACS data are available at https://data.nodc.noaa.gov/cgi-bin/iso?id=gov.noaa.ncdc:C01552. The NCEP-DOE reanalysis dataset is available at https://www.esrl.noaa.gov/psd/data/gridded/data.ncep.reanalysis2.html. The large-ensemble and CMIP5 datasets are available at: http://www.cesm.ucar.edu/projects/community-projects/LENS/ (LENS), https://esgf-data.dkrz.de/search/mpi-ge/ (MPI-GE), https://www.earthsystemgrid.org/dataset/ucar.cgd.ccsm4.CLIVAR_LE.gfdl_cm3_lens.html (GFDL-

LE), and https://esgf-node.llnl.gov/projects/cmip5/ (CMIP5). The downscaling data used in Fig. 4 and Supplementary Fig. 17 are provided as Supplementary Data files.

## Code availability
All computer codes used to analyze the data and produce the plots are available from the corresponding author upon request.

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

## Acknowledgements

We thank Phil Bedient, Dan Cohan, Noah Diffenbaugh, Kerry Emanuel, Melissa Gervais, Sandro Lubis, John Nielsen-Gammon, Morgan O'Neill, Toni Sebastian, and Robb Wills for fruitful discussions. This work is supported by NSF grant AGS-1921413, NASA grant 80NSSC17K0266, and an Early-Career Research Fellowship from the Gulf Research Program of the National Academies of Sciences, Engineering, and Medicine (to P.H.), Rice Houston Engagement and Recovery Effort Fund (to P.H. and L.Y.Y.), a Columbia Climate and Life Fellowship, CCL (to C.-Y.L.), and NOAA grants NA16OAR4310079 and NA18OAR4310277 (to S.J.C.). The CHAZ data were generated under a research project supported by the New York State Energy Research and Development Authority (NYSERDA 103862). Computational resources were provided by XSEDE (allocation ATM170020), NCAR's CISL (allocation URIC0004), and Rice University Center for Research Computing.

## Author contributions

P.H. designed and supervised the study. P.H., C.-Y.L., E.N., and L.Y.Y. developed codes and analyzed data. P.H. and C.-Y.L. wrote the manuscript. P.H., C.-Y.L., E.N., S.J.C., D.M., and L.Y.Y. interpreted the results, discussed the findings, and revised the manuscript.

## Competing interests

The authors declare no competing interests.
