## [Peer Review File · Nature Communications]

Peer Review File - Reviewers' comments:

Reviewer #1 (Remarks to the Author):

Overall Recommendation: Minor revisions needed before acceptance.

This paper reads very well and has an interesting approach to understanding future Texas hurricane translation speed. The results are novel and offer a differing perspective than what is in the literature. The statistical and numerical modeling approaches appear sound and are well explained in the supplement.

I appreciate the authors' discussion of other studies similar/different to theirs and how it fits into current literature. I feel the biggest flaw is the lack of discussion on how someone might actually use this information. What do the results mean for people who are actually planning for mitigation efforts in the state? I think the discussion of Ike vs. Harvey in the conclusions (starting in line 267) needs to be expanded upon. If both resulted in intense damage amounts from differing factors, then why does it matter that we know if the storms are slowing or moving more quickly? Can you speak to this? Bringing in other events (i.e., Michael vs. Florence) might help to make your case, in my opinion. In addition, can you speak to whether we would expect similar changes to landfalling events in other places within the Gulf of Mexico: Louisiana or Florida, for example, based on what you found in your results?

Small editorial changes are needed – minor edits to sentence structure (missing “the” in a few places)

Lines 49-50: Add in estimated cost amounts for the three storms listed.

Reviewed by:
Jill Trepanier
Assoc. Prof.
LSU

Reviewer #2 (Remarks to the Author):

Review of: Effects of climate change on the movement of future landfalling Texas tropical cyclones
by Hassanzadeh et al. 2020.

Submitted to Nature, Feb. 2020.

I have studied the submitted manuscript and find the overall study to be a plausible analysis of the steering flow changes affecting late 21st century tropical cyclones over the Texas region of the United States. As the authors correctly state in their Conclusions: This work will "help disentangle the contributions from changes in dynamics (large-scale circulation) and thermodynamics (changes in temperature) ... on the risk of TCs making landfall in Texas."

The authors conduct several diagnostic analyses of changes in the steering flow using an ensemble of model simulations under the RCP8.5 scenario. The authors find that the northward component of the steering flow over Texas in this hypothetical (model) climate contributes to an increased likelihood of fast-moving tropical cyclones around the Houston area. The authors attribute these changes in the steering flow to changes in the Atlantic subtropical high and American monsoon flows during the June-September time frame. The authors suggest that their findings are not dissimilar to some recent work by Yamaguchi et al., but differ from that of Gutmann et al. The authors point out that the different conclusions reached by themselves and Guntmann appear to

be due to major differences in the methodologies employed to quantify TC track- changes in future climate flows. The authors recommend further analysis of the difference between their findings and those of Guntmann.

Despite my overall positive assessment, I think the mss. can be improved without too much major revision.

1. I would like the authors to provide some additional justification for their choice of using the RCP8.5 scenario. As an example, the NOAA GFDL group led by Knutson et al. 2015 focused on the more moderate RCP4.5 scenario for their storm intensity and storm frequency assessments.

> Careful justification for the use of RCP8.5 needs to be provided in the main text.

2. The work presented here offers new evidence that Hurricane Harvey-like events over Texas will be less likely near the end of the 21st century in this model scenario. On its face, this study offers new model evidence to counter some recent claims that Harvey-like events will become much more likely near the end of 21st century.

> Are the authors 100% satisfied with their explanation of their findings? I am not 100% convinced of the proffered explanation. In particular, why does the subtropical high near the Texas coast and Gulf region and the north american monsoon shift zonally and strengthen during this hypothetical climate?

3. While reading the mss. carefully, I came across a few minor grammatical mistakes.

> I recommend carefully reading the revised mss. so that missing articles, etc. are corrected.

e.g.:

l161: ... indicates a weakening of such winds ...

l178: changes in the translation speed ...

l247: midlatitudes ...

Recommendation: Accept with some revision (noted above).

Future work: To assess the robustness of the results, I recommend extending this study to include the more moderate RCP4.5 climate change scenario as conducted by NOAA GFDL (e.g., Knutson et al. 2015).

Reviewer #3 (Remarks to the Author):

Review of NCOMMS-20-03454-T, Hassanzadeh et al, "Effects of climate change on the movement of future landfalling Texas tropical cyclones"

Recommendation: Revision.

My concerns primarily address clarity of explanation. In cases where I have specific questions about methodology, I am assuming details were omitted rather than not performed. If the latter holds, it's possible that results/conclusions could change but I consider that unlikely since the methods are fairly robust.

Summary

As noted below, my primary focus is on the statistical analysis (the SOMs) but I did still wish to comment on the overall manuscript from the perspective of general climate and atmospheric science. Even as a non-specialist, I found the manuscript interesting scientifically and at least qualitatively convincing. It is well-motivated and organized well to support the analysis results. The overall analysis is designed to address important aspects of the questions being asked (e.g., how do the case studies differ? how will key aspects of the climate change in future projections, at large- and regional scales?). The authors apply the model ensembles very effectively to the research questions. And I believe the conclusions are supported by the analysis.

Note that I am not commenting on the thoroughness of the "literature review" portion or whether prior results are presented appropriately as these are outside my expertise. Likewise for the hazards modeling.

SOMs

I was asked specifically to comment on the statistical analysis aspects of the manuscript. I have not used this particular implementation (the Matlab Deep Learning Toolbox, which should be mentioned more clearly), but the authors appear to have applied it sensibly. I did have a few specific questions/concerns though:

1. Future vs. current (Main, lines 144-170; Data and Methods, lines 82-101)

a) I am concerned about some terminology: patterns shown in the SOM figures are "cluster centers" (the average of the daily patterns within each cluster) rather than the generalized SOM pattern itself – and this is why the "patterns can change" under climate change (lines 152-153). It's possible that it's because I'm just so used to seeing the SOM patterns themselves that I found this usage so confusing. Or maybe more needs to be said explicitly about how the patterns in Fig S13 are an averaged version of data variability in the FULL current-plus-future dataset, i.e., it's not quite enough to just say that the SOM analyzed the combined dataset. The reader ought to be reminded that the temporal subsets (current, future) will yield different frequencies AND cluster centers. Put another way, unless the reader is fully aware that the patterns (cluster centers) shown come from the FULL dataset, it may not be clear why they might change over time.

b) The combined dataset analysis also means that the frequencies shown in S13 are for the combined dataset, which is arguably not as meaningful as showing the frequencies for current and future separately (although you do partly do this in Fig 3 with current frequency and change-in-frequency). This is handled properly in the analyses based on equations 3-5, where future/current frequencies appear explicitly, but either the S13 titles or caption could be changed to clarify which frequencies are being shown.

c) Are generalized SOM patterns are presented in any of the figures? I thought maybe in S11 and S12 but the captions are not clear. It's not that you have to show them, just that some captions may need editing for clarification.

2. Data preprocessing

Was any normalization of the data done prior to running the SOM algorithm? With only one variable ("winds") this is less critical than when variables with different means and variance are done together, but it may still be relevant if winds have notably different statistics either spatially or between time periods. Data preprocessing is partly covered (Data and Methods, lines 58-60) but it's unclear whether this mentions all steps taken. Please address/clarify this aspect.

3. LENS vs reanalysis (Figs S11, S12)

Was this a single SOM analysis or are two different SOMs being compared (one for reanalysis, one for LENS)? If this is single SOM (reanalysis plus LENS), what, if anything, was done to keep the LENS data from swamping the reanalysis data (40 models to one, essentially)? Judging by the figures, I assume that two distinct SOMs were used and, based on S12's caption, intercomparison was done simply by ordering the patterns by frequency? This strategy mostly works at a rather subjective level (though I would not want to base any further conclusions on the bottom row). Since the authors don't really go beyond a subjective comparison (Main, lines 148-150; Data and Methods, lines 72-74), I'm ok with this. Ideally, though, something more rigorous would be better (especially if this was a LENS vs reanalysis manuscript). Please clarify exactly how the SOM was

applied in the different dataset analyses (following the good example at Data and Methods lines 82-84).

Minor Comments

4. Data and Methods, Line 84: Should "S12" be "S13"?

5. Figs 3, S11-16: At the risk of adding distracting lines, I suspect international readers would benefit from at least one panel having national/subnational boundaries in these maps.

6. Abstract: lines 38-39, perhaps more accurate to say "a 10 percentage point shift"?

Reviewer #1

Reviewer:

Overall Recommendation: Minor revisions needed before acceptance.

This paper reads very well and has an interesting approach to understanding future Texas hurricane translation speed. The results are novel and offer a differing perspective than what is in the literature. The statistical and numerical modeling approaches appear sound and are well explained in the supplement.

Authors:

Dear Professor Trepanier,

We thank you for your careful read of our paper and for providing insightful and helpful comments/suggestions. We have addressed your comments/suggestions by revising the text and we have provided point-by-point responses below. A version of the revised manuscript with changes tracked is attached to the end of the response.

Reviewer:

I appreciate the authors' discussion of other studies similar/different to theirs and how it fits into current literature. I feel the biggest flaw is the lack of discussion on how someone might actually use this information. What do the results mean for people who are actually planning for mitigation efforts in the state? I think the discussion of Ike vs. Harvey in the conclusions (starting in line 267) needs to be expanded upon. If both resulted in intense damage amounts from differing factors, then why does it matter that we know if the storms are slowing or moving more quickly? Can you speak to this? Bringing in other events (i.e., Michael vs. Florence) might help to make your case, in my opinion. In addition, can you speak to whether we would expect similar changes to landfalling events in other places within the Gulf of Mexico: Louisiana or Florida, for example, based on what you found in your results?

Authors:

Thank you for these questions and for your suggestions. We agree with you that this issue deserves further clarification. We have added a few sentences and references (Lines 306-312) to explain why it matters to know whether the future TCs will move faster or slower and how the main drivers of damage will change:

“However, as mentioned earlier, the damages were due to differing factors: Intense rainfall (which led to extensive flooding) in the case of Harvey, and storm surge-induced flooding and wind gust in the case of Ike. Understanding the main driver(s) of damage by future TCs in each region is crucial for adaptation and mitigation efforts, as different drivers require different—and often costly and controversial—protective measures and strategies (e.g., seawalls for storm surge vs. improved reservoirs and bayou systems for rainfall-induced flooding)^{35,36,37,38,39,40,41}.”

Adaptation and mitigation efforts can be very costly, time consuming, and controversial, for Houston

<https://www.houstonchronicle.com/news/houston-texas/houston/article/Ike-Dike-concept-raises-more-questions-than-it-13378121.php>

<https://www.houstonchronicle.com/news/houston-texas/houston/article/H0826-Harvey-dams-levees-13180458.php>

and for other regions

<https://www.nytimes.com/2020/01/17/nyregion/sea-wall-nyc.html>

and uncertainties in projections of future TCs' risk complicates the decision-making process of these crucial efforts. As further discussed in Lines 314-327, analyses such as ours are essential for better understanding and constraining the future changes.

As we emphasize in the revised text (Lines 249-253), our work suggests the need for studying each region separately, using a multi-model, multi-faceted approach. Thus, for this paper, we aim to keep the focus on Texas, but we totally agree with the need to look into other events such as Michael, Florence, Dorian etc. in other regions. We leave such analyses to future work. That said, from examining the results of Figs. 2, 3, and S1, we can see faster northward steering winds over Louisiana too, and thus expect an increase in the likelihood of faster-moving TCs making landfall in Louisiana under climate change. However, to further confirm this, the CMIP5-TC downscaling experiments focused on Louisiana are needed, which we leave to future work.

The question about Florida is very interesting. From Figs. 1 and S1, the projections of steering winds over Florida are highly uncertain. Given such complexities, the multi-model, multi-faceted approach is certainly even more critical for Florida. This is left for future work.

Again, thank you for bringing these issues to our attention.

Reviewer:

Small editorial changes are needed – minor edits to sentence structure (missing “the” in a few places)

Authors:

Thank you for pointing out this issue. We have proofread the revised manuscript carefully to fix these issues.

Reviewer:

Lines 49-50: Add in estimated cost amounts for the three storms listed.

Authors:

Thank you for the suggestion. We have added the estimated costs for the three storms (Line 52).

Reviewer #2

Reviewer:

I have studied the submitted manuscript and find the overall study to be a plausible analysis of the steering flow changes affecting late 21st century tropical cyclones over the Texas region of the United States. As the authors correctly state in their Conclusions: This work will "help disentangle the contributions from changes in dynamics (large-scale circulation) and \ thermodynamics (changes in temperature) ... on the risk of TCs making landfall in Texas."

The authors conduct several diagnostic analyses of changes in the steering flow using an ensemble of model simulations under the RCP8.5 scenario. The authors find that the northward component of the steering flow over Texas in this hypothetical (model) climate contributes to an increased likelihood of fast-moving tropical cyclones around the Houston area. The authors attribute these changes in the steering flow to changes in the Atlantic subtropical high and American monsoon flows during the June-September time frame. The authors suggest that their findings are not dissimilar to some recent work by Yamaguchi et al., but differ from that of Guntmann et al. The authors point out that the different conclusions reached by themselves and Guntmann appear to be due to major differences in the methodologies employed to quantify TC track- changes in future climate flows. The authors recommend further analysis of the difference between their findings and those of Guntmann.

Despite my overall positive assessment, I think the mss. can be improved without too much major revision.

Authors:

We thank the reviewer for their careful read of our paper and for providing insightful and helpful comments/suggestions. We have addressed their comments/suggestions by revising the text and we have provided point-by-point responses below. A version of the revised manuscript with changes tracked is attached to the end of the response.

Reviewer:

1. I would like the authors to provide some additional justification for their choice of using the RCP8.5 scenario. As an example, the NOAA GFDL group led by Knutson et al. 2015 focused on the more moderate RCP4.5 scenario for their storm intensity and storm frequency assessments. > Careful justification for the use of RCP8.5 needs to be provided in the main text.

Authors:

We appreciate that you raised this point. The RCP8.5 is used in most other studies of changes in TC movement and changes in large-scale circulation under climate change. Using RCP8.5 enables us to compare our results with those reported in other studies. We have also focused, like most other studies, on changes by the end of the 21st century. RCP8.5 and the end of 21st century are widely used, as they give the largest responses. We do not expect RCP4.5 to lead to new/unexpected responses compared to RCP8.5, but just to weaker responses that might emerge

at later times. But, we totally agree that to best inform the adaptation and mitigation efforts and policy making, projections from the mid-21st century and using other scenarios such as RCP4.5 are needed too. We have left such analyses to future work, but added the following sentences to the end of the Discussion section (Lines 322-326)

“Finally, in this paper, as in most other studies, we focus on changes in the late 21st century under the high-emission scenario, RCP8.5 (thus allowing comparison with previously reported results). However, to better inform the adaptation and mitigation efforts, mainly about the time of emergence and the magnitude of these changes in TC movement, similar analyses for the mid-21st century and under other emission scenarios such as RCP4.5 (as e.g., discussed in Knutson et al.^{11,14}) should be conducted in future work.”

Reviewer:

2. The work presented here offers new evidence that Hurricane Harvey-like events over Texas will be less likely near the end of the 21st century in this model scenario. On its face, this study offers new model evidence to counter some recent claims that Harvey-like events will become much more likely near the end of 21st century.

> Are the authors 100% satisfied with their explanation of their findings? I am not 100% convinced of the proffered explanation. In particular, why does the subtropical high near the Texas coast and Gulf region and the north american monsoon shift zonally and strengthen during this hypothetical climate?

Authors:

We are very confident in our explanation *that the robust future increase in northward steering wind over Texas is due to the constructive influence of a west-ward shifting and intensifying Atlantic subtropical high and a weakening American monsoon.* We would like to further clarify a few points:

- 1- That the Atlantic subtropical high will intensify and shift westward under climate change has been reported in previous studies, e.g., in Li et al. (<https://doi.org/10.1038/ngeo1590>). They attributed these changes to the enhanced land-ocean thermal contrast. In our paper, we have further shown that these changes are robust in 3 different sets of large-ensemble simulations too. Further examination of the underlying dynamics of these changes in the Atlantic subtropical high is certainly an important issue, but beyond the scope of our paper. We have cited the work of Li et al. and mentioned the role of land-ocean thermal contrast.
- 2- That the American monsoon will weaken under climate change has been reported in previous studies, e.g., in Pascale et al. (<https://doi.org/10.1038/nclimate3412>). Weakening of the North American monsoon is associated with reduced monsoonal precipitation and diabatic heating in the upper troposphere. Through Gill model (Gill, Quart. J. R. Met. SOC. (1980). 106, pp. 447-462), this change in the North American monsoon leads to robust anomalous upper-level northward winds over Texas. Pascale et al. proposed that the weakening of the North American monsoon can be attributed to increased atmospheric stability due to uniform sea-surface warming. Again, in our paper we have shown that the weakening of the American monsoon is robust in 3 different sets of large-ensemble

simulations too. But, further examination of the underlying dynamics, while certainly an important issue, is beyond the scope of our paper. In the revised paper, we have cited the work of Pascale et al. and Gill and mentioned the role of increased static stability.

- 3- Here is what is new in our paper regarding this topic: To the best of our knowledge, this is the first study that examines the TC steering winds over Texas, and finds the connections with changes in Atlantic subtropical high and American monsoon. We were surprised that we found such robust regional response in circulation, and that encouraged us to dig more into the underlying mechanism (note that our explanation builds on the recent work by Wills et al. <https://doi.org/10.1007/s40641-019-00147-6>). As discussed above, the changes in the subtropical high and American monsoon have been known for a while, but their impact on the circulation over Texas, and certainly their constructive influence, had not been noticed before.

In summary, we are very confident about our explanation for why there is a robust regional response in TC steering winds. But admittedly, further work is needed to ensure that the current understanding of why the subtropical high intensifies and shifts, and why the American monsoon weakens, are 100% complete. Such work is beyond the scope of this paper, but certainly should be pursued in the future.

Also, along the lines of the reviewer's previous comments, we are now inspired to think more about how these changes in the subtropical high and American monsoon depend on the radiative forcing scenario (RCP8.5 vs RCP4.5) and time period (mid 21st century vs the end of the 21st century). Whether these changes emerge at the same time, and how they are impacted by the forcing scenario, are important questions that require a systematic, hierarchical modeling approach. We aim to address these questions in future work.

We thank the reviewer for this insightful question.

Reviewer:

3. While reading the mss. carefully, I came across a few minor grammatical mistakes.
> I recommend carefully reading the revised mss. so that missing articles, etc. are corrected.
e.g.:
1161: ... indicates a weakening of such winds ...
1178: changes in the translation speed ...
1247: midlatitudes ...

Authors:

Thank you for pointing out these typos. We have carefully proofread the revised manuscript.

Reviewer:

Recommendation: Accept with some revision (noted above).

Authors:

We thank the reviewer for insightful comments and suggestions.

Reviewer:

Future work: To assess the robustness of the results, I recommend extending this study to include the more moderate RCP4.5 climate change scenario as conducted by NOAA GFDL (e.g., Knutson et al. 2015).

Authors:

We totally agree with you. As mentioned above, we have added sentences to the Discussion emphasizing the need for such analysis in future work. We thank the reviewer for their thoughtful recommendation.

Reviewer #3

Reviewer:

Recommendation: Revision.

My concerns primarily address clarity of explanation. In cases where I have specific questions about methodology, I am assuming details were omitted rather than not performed. If the latter holds, it's possible that results/conclusions could change but I consider that unlikely since the methods are fairly robust.

Summary

As noted below, my primary focus is on the statistical analysis (the SOMs) but I did still wish to comment on the overall manuscript from the perspective of general climate and atmospheric science. Even as a non-specialist, I found the manuscript interesting scientifically and at least qualitatively convincing. It is well-motivated and organized well to support the analysis results. The overall analysis is designed to address important aspects of the questions being asked (e.g., how do the case studies differ? how will key aspects of the climate change in future projections, at large- and regional scales?). The authors apply the model ensembles very effectively to the research questions. And I believe the conclusions are supported by the analysis.

Note that I am not commenting on the thoroughness of the "literature review" portion or whether prior results are presented appropriately as these are outside my expertise. Likewise for the hazards modeling.

Authors:

We thank the reviewer for their careful read of our paper and for providing insightful and helpful comments/suggestions, particularly to clarify our clustering methodology and presentation of results. We have addressed their comments/suggestions by revising the text (and in some cases figures) and we have provided point-by-point responses below. A version of the revised manuscript with changes tracked is attached to the end of the response.

We would like to point out that we have extensively tested the robustness of our cluster analysis with respect to parameter choices such as the number of clusters, clustering methodology (e.g., using K-means clustering in additions to SOMs), and approach to current/future clusters comparison (e.g., by clustering the current and future climates separately or together). *We always reached the same conclusion.* The point is that the increase in northward steering winds is such a strong signal that it robustly emerges regardless of the details of the analysis.

Reviewer:

SOMs

I was asked specifically to comment on the statistical analysis aspects of the manuscript. I have not used this particular implementation (the Matlab Deep Learning Toolbox, which should be

mentioned more clearly), but the authors appear to have applied it sensibly. I did have a few specific questions/concerns though:

1. Future vs. current (Main, lines 144-170; Data and Methods, lines 82-101)

a) I am concerned about some terminology: patterns shown in the SOM figures are “cluster centers” (the average of the daily patterns within each cluster) rather than the generalized SOM pattern itself – and this is why the “patterns can change” under climate change (lines 152-153). It’s possible that it’s because I’m just so used to seeing the SOM patterns themselves that I found this usage so confusing. Or maybe more needs to be said explicitly about how the patterns in Fig S13 are an averaged version of data variability in the FULL current-plus-future dataset, i.e., it’s not quite enough to just say that the SOM analyzed the combined dataset. The reader ought to be reminded that the temporal subsets (current, future) will yield different frequencies AND cluster centers. Put another way, unless the reader is fully aware that the patterns (cluster centers) shown come from the FULL dataset, it may not be clear why they might change over time.

Authors:

We agree with the reviewer that further clarification of this approach was needed. We would like to point out that in earlier stages of this work, we clustered the current and future climates *separately*. From inspecting the clusters, we found that clusters that steer hurricanes northward become more prevalent under climate change. Later, we came across the work of Gervais et al. (2020, in press, <https://doi.org/10.1175/JCLI-D-19-0636.1>), who, to understand local changes in the midlatitude jet under climate change, clustered the current and future climates together. While both approaches make sense, and in our case, lead to the same conclusion, we adopted the approach of Gervais et al. because it is more quantitative, and allows us (through Eqs. 3-5) to quantify and separate the contributions from changes in frequency and in pattern. Furthermore, this approach allows us to seamlessly connect the cluster analysis with the June-September-averaged steering wind analysis (Fig. 3n is the same as the combination of the responses in Figs. 2a and 2c).

Again, we agree that further clarification was needed. We have revised the Main text (Lines 172-182) and Methods (Lines 412-416) to ensure that the methodology is clear to the readers. We also mentioned the use of MATLAB’s Deep Learning Toolbox in Data and Methods (Line 385-386).

Reviewer:

b) The combined dataset analysis also means that the frequencies shown in S13 are for the combined dataset, which is arguably not as meaningful as showing the frequencies for current and future separately (although you do partly do this in Fig 3 with current frequency and change-in-frequency). This is handled properly in the analyses based on equations 3-5, where future/current frequencies appear explicitly, but either the S13 titles or caption could be changed to clarify which frequencies are being shown.

Authors:

Thank you for raising this point. As you mentioned, the frequencies are handled properly in the analyses through Eqs. (3)-(5), which is a strength of the approach taken here. We agree that further clarification, particularly in the captions, was needed. Therefore, we have made the following changes

- In Methods (Lines 412-416) and in the caption of Fig. 3, now we explicitly define f_i^C , f_i^F , and \bar{f}_i : For each cluster i , the frequency f_i^C (f_i^F) is defined as the number of days in the current (future) period in that cluster divided by the total number of days in the current (= future) period. $\bar{f}_i = (f_i^C + f_i^F)/2$.

- In the caption of Figs. 3 and S13-S14, we explicitly and clearly mention what frequency each number is representing. Please note that while we are not reporting f_i^C or f_i^F separately, Figs. 3 and S13-S14 present the mean (\bar{f}_i) and difference of these two numbers for each cluster. An interested reader can then easily compute f_i^C and f_i^F for each cluster if they wish.

Reviewer:

c) Are generalized SOM patterns are presented in any of the figures? I thought maybe in S11 and S12 but the captions are not clear. It's not that you have to show them, just that some captions may need editing for clarification.

Authors:

Yes, Figs. S11 and S12 show the SOM patterns of the reanalysis data and of the LENS current climate separately (as discussed in the response to Q3, two separate SOM analyses were conducted to obtain the results shown in Figs. S11 and S12). We have revised the captions of both figures to clarify this.

Reviewer:

2. Data preprocessing

Was any normalization of the data done prior to running the SOM algorithm? With only one variable ("winds") this is less critical than when variables with different means and variance are done together, but it may still be relevant if winds have notably different statistics either spatially or between time periods. Data preprocessing is partly covered (Data and Methods, lines 58-60) but it's unclear whether this mentions all steps taken. Please address/clarify this aspect.

Authors:

Thanks for raising this issue. Given that we are looking at only wind vectors and that they are overall from limited spatial and temporal windows, and to avoid complications coming from pre-processing, we have not done any pre-processing of the data before conducting the SOM analysis. We have added a sentence (Line 384-385 in Methods) to clarify this.

Reviewer:

3. LENS vs reanalysis (Figs S11, S12)

Was this a single SOM analysis or are two different SOMs being compared (one for reanalysis, one for LENS)? If this is single SOM (reanalysis plus LENS), what, if anything, was done to keep the LENS data from swamping the reanalysis data (40 models to one, essentially)? Judging by the figures, I assume that two distinct SOMs were used and, based on S12's caption, intercomparison was done simply by ordering the patterns by frequency? This strategy mostly works at a rather subjective level (though I would not want to base any further conclusions on the bottom row).

Since the authors don't really go beyond a subjective comparison (Main, lines 148-150; Data and Methods, lines 72-74), I'm ok with this. Ideally, though, something more rigorous would be better (especially if this was a LENS vs reanalysis manuscript). Please clarify exactly how the SOM was applied in the different dataset analyses (following the good example at Data and Methods lines 82-84).

Authors:

We appreciate this comment and the suggestions, which help with clarifying our approach.

- Yes, we conducted two separate SOM analyses to obtain these results: One applied to the reanalysis data (Fig. S11) and the other applied to the LENS current climate data (Fig. S12). We have now clarified this in the captions and in Methods (Lines 398-401).

- Yes, we just found the corresponding clusters based on pattern correlation and then compared the frequencies and visually inspected the patterns, and we found a qualitative agreement. In fact, we find the agreement pretty remarkable, given that this is a comparison of regional, daily variability between reanalysis and a climate model. We did not find GFDL-LE to work as well (mainly missing one of the clusters). *Still, to be quantitative, we have added the pattern correlations and errors in frequencies to the caption of Fig. S12.*

Reviewer:

Minor Comments

4. Data and Methods, Line 84: Should "S12" be "S13"?

Authors:

Yes, you are right. Thank you for catching this. We have carefully read the revised Main text and Methods to ensure that they are typo free.

Reviewer:

5. Figs 3, S11-16: At the risk of adding distracting lines, I suspect international readers would benefit from at least one panel having national/subnational boundaries in these maps.

Authors:

Thanks for the suggestion. We added the international and state borders to Figs. 3 and S11-S16.

Reviewer:

6. Abstract: lines 38-39, perhaps more accurate to say "a 10 percentage point shift"?

Authors:

Thanks for the suggestion. Done.

REVIEWERS' COMMENTS:

Reviewer #1 (Remarks to the Author):

Thank you for addressing my concerns. I now feel it is ready for publication.

Reviewer #2 (Remarks to the Author):

2nd review of: "Effects of climate change on the movement of future landfalling Texas tropical cyclones"

Summary and Evaluation:

I have studied carefully the authors' response to my prior review. The authors have constructed a thoughtful response to my comments and have done a very fine job revising their manuscript. I think the authors have responded constructively to the other reviewers also. Overall, I think the revised manuscript is much improved and is now ready for prime time.

In summary, the authors' revised manuscript provides a plausible analysis of projected climate change impacts on future landfalling Texas tropical cyclones using the RCP8.5 scenario.

Although the authors present ample evidence of the robustness of their findings and interpretation, they acknowledge also some possible limitations of the RCP8.5 scenario as a basis for future projections and they acknowledge differences with other studies adopting a more global focus.

Recommendation: Accept.

I have no problem waiving the anonymity of my review at this time.

Michael T. Montgomery
Naval Postgraduate School
Monterey CA 93943

Reviewer #3 (Remarks to the Author):

Review of NCOMMS-20-03454A, Hassanzadeh et al, "Effects of climate change on the movement of future landfalling Texas tropical cyclones"

Recommendation: Accept.

The responses to my prior comments are entirely satisfactory and the authors have made numerous improving changes to the manuscript. I thank them for their efforts in making these revisions and look forward to future publications on this topic.

Reviewer #1:

Thank you for addressing my concerns. I now feel it is ready for publication.

Authors:

Dear Professor Trepanier,

We thank you for your earlier comments and suggestions, which substantially improved the manuscript.

Reviewer #2:

2nd review of: "Effects of climate change on the movement of future landfalling Texas tropical cyclones"

Summary and Evaluation:

I have studied carefully the authors' response to my prior review. The authors have constructed a thoughtful response to my comments and have done a very fine job revising their manuscript. I think the authors have responded constructively to the other reviewers also. Overall, I think the revised manuscript is much improved and is now ready for prime time.

In summary, the authors' revised manuscript provides a plausible analysis of projected climate change impacts on future landfalling Texas tropical cyclones using the RCP8.5 scenario.

Although the authors present ample evidence of the robustness of their findings and interpretation, they acknowledge also some possible limitations of the RCP8.5 scenario as a basis for future projections and they acknowledge differences with other studies adopting a more global focus.

Recommendation: Accept.

I have no problem waiving the anonymity of my review at this time.

Michael T. Montgomery
Naval Postgraduate School, Monterey CA 93943

Authors:

Dear Professor Montgomery,

We thank you for your earlier comments and suggestions, which substantially improved the manuscript.

Reviewer #3:

Review of NCOMMS-20-03454A, Hassanzadeh et al, "Effects of climate change on the movement of future landfalling Texas tropical cyclones"

Recommendation: Accept.

The responses to my prior comments are entirely satisfactory and the authors have made numerous improving changes to the manuscript. I thank them for their efforts in making these revisions and look forward to future publications on this topic.

Authors:

We thank the reviewer for their earlier comments and suggestions, which substantially improved the manuscript.